# Wnt signaling-mediated redox regulation maintains the germ line stem cell differentiation niche

Su Wang[1,2], Yuan Gao[3], Xiaoqing Song[1], Xing Ma[1,2], Xiujuan Zhu[1], Ying Mao[3], Zhihao Yang[3], Jianquan Ni[3], Hua Li[1], Kathryn E Malanowski[1], Perera Anoja[1], Jungeun Park[1], Jeff Haug[1], Ting Xie[1,2]*

[1]Stowers Institute for Medical Research, Kansas City, United States; [2]Department of Anatomy and Cell Biology, University of Kansas School of Medicine, Kansas City, United States; [3]Center for Life Sciences, College of Life Sciences, School of Medical Sciences, Tsinghua University, Beijing, China

**Abstract** Adult stem cells continuously undergo self-renewal and generate differentiated cells. In the *Drosophila* ovary, two separate niches control germ line stem cell (GSC) self-renewal and differentiation processes. Compared to the self-renewing niche, relatively little is known about the maintenance and function of the differentiation niche. In this study, we show that the cellular redox state regulated by Wnt signaling is critical for the maintenance and function of the differentiation niche to promote GSC progeny differentiation. Defective Wnt signaling causes the loss of the differentiation niche and the upregulated BMP signaling in differentiated GSC progeny, thereby disrupting germ cell differentiation. Mechanistically, Wnt signaling controls the expression of multiple *glutathione-S-transferase* family genes and the cellular redox state. Finally, *Wnt2* and *Wnt4* function redundantly to maintain active Wnt signaling in the differentiation niche. Therefore, this study has revealed a novel strategy for Wnt signaling in regulating the cellular redox state and maintaining the differentiation niche.

*For correspondence:
tgx@stowers.org

Competing interests: The authors declare that no competing interests exist.

## Introduction

Stem cells have two important properties, self-renewal and differentiation, which are critical for continuously generating new functional cells to maintain tissue homeostasis. The self-renewal property is controlled in various stem cell systems by interplays between signals from the niche and intrinsic factors (*Li and Xie, 2005*; *Morrison and Spradling, 2008*; *Losick et al., 2011*). Germ line stem cells (GSCs) in the *Drosophila* ovary and testis are attractive systems for studying stem cell self-renewal at the molecular and cellular level (*Fuller and Spradling, 2007*; *Xie, 2013*). Although stem cell differentiation was widely thought to be a developmentally default state, we have recently proposed that GSC lineage differentiation is also controlled extrinsically by a differentiation niche formed by inner germarial sheath cells (ISCs, also known as escort cells). However, it remains unclear how the maintenance and function of the differentiation niche are regulated at the molecular level. In this study, we show that autocrine Wnt2/4 signaling maintains the differentiation niche by regulating ISC proliferation and survival via redox regulation.

In the *Drosophila* ovary, two or three GSCs at the tip of the germarium, the most anterior region of the *Drosophila* ovary, continuously self-renew and generate differentiated GSC daughters, cystoblasts (CBs). The CBs further divide four times synchronously with incomplete cytokinesis to form 2-cell, 4-cell, 8-cell, or 16-cell cysts (*de Cuevas et al., 1997*). GSCs and their differentiated progeny can be reliably identified by their unique morphology of germ line-specific intracellular organelles known as fusomes:

**eLife digest** An animal or plant has many different types of cells that have specific roles in the life of the organism. These cells are organized into tissues. In most tissues in adult animals, small groups of cells called stem cells are responsible for replacing the other cells that have been lost due to disease, injury, or as part of normal body maintenance.

The 'germ line' stem cells of female fruit flies—which produce female sex cells (or eggs)—are an effective system for studying how stem cells are regulated. These cells live in an area of the ovary called a stem cell niche. Each time a stem cell divides, it produces one stem cell and one other daughter cell. This daughter cell then moves into another niche called the 'differentiation' niche and undergoes a series of divisions that produce the egg cells. The differentiation niche is formed by escort cells and is crucial for producing the egg cells, but it is not clear how the escort cells promote this process, or how the niche is maintained.

Wang et al. have now studied the differentiation niche in more detail. The experiments show that a cell communication system called Wnt signaling maintains the differentiation niche by controlling the ability of the escort cells to grow and divide. If Wnt signaling is defective, the differentiation niche is lost, which disrupts the formation of egg cells.

Further experiments show that two proteins called Wnt2 and Wnt4 in the differentiation niche—which activate Wnt signaling—act as signals to regulate the niche, mainly by controlling the expression of four particular genes. These four genes encode enzymes that remove 'reactive oxygen species' from cells. Wang et al.'s findings have revealed an important role for Wnt signaling in maintaining the differentiation niche. The next step is to figure out the details of how this works.

GSCs and CBs contain a spherical fusome known as the spectrosome, whereas differentiated germ cell cysts contain a branched fusome (*Lin et al., 1994*). GSCs can be reliably distinguished from CBs by their direct contact with cap cells (*Figure 1A*). Cap cells function as the self-renewing niche to maintain GSCs by activating BMP signaling and maintaining E-cadherin-mediated cell adhesion (*Song et al., 2002*; *Xie and Spradling, 1998*, *2000*). In addition, various classes of intrinsic factors work with BMP signaling and E-cadherin to control GSC self-renewal (*Xie, 2013*). Therefore, GSC self-renewal is controlled by coordinated functions of niche-initiated signaling pathways and intrinsic factors.

Following GSC division, differentiating GSC daughters, CBs, are always positioned away from the self-renewal niche. ISCs sit on the surface of the germarium to send their cellular processes to wrap up underneath CBs, mitotic cysts, and early 16-cell cysts, which move posteriorly (*Decotto and Spradling, 2005*; *Kirilly et al., 2011*; *Morris and Spradling, 2011*). Our recent study suggests ISCs and their associate long cellular processes act as the differentiation niche to promote GSC progeny differentiation in the *Drosophila* ovary because disrupting long ISC processes leads to an accumulation of CB-like cells, indicative of a germ cell differentiation defect (*Kirilly et al., 2011*). A series of genetic studies have further supported the existence of the differentiation niche.

The epidermal growth factor (EGF) signaling pathway is active in ISCs to promote GSC lineage differentiation partly by repressing *dally* expression (*Schultz et al., 2002*; *Liu et al., 2010*). In addition, Rho signaling is also required in ISCs to promote GSC differentiation partly by repressing *dally* and *dpp* expression. *dally* encodes a proteoglycan protein, which is capable of promoting Dpp/BMP diffusion to the differentiation niche (*Guo and Wang, 2009*; *Hayashi et al., 2009*). Ecdysteroid signaling also operates in ISCs to promote germ cell differentiation because inactivating ecdysteroid receptors EcR and Usp in ISCs disrupts cyst formation (*Morris and Spradling, 2012*). One potential mechanism is that ecdysteroid signaling controls the formation of ISC cellular processes, thereby promoting the interaction between ISCs and germ cells (*Konig and Shcherbata, 2015*). Gap junction protein Inx2 functions in ISCs to promote germ cell differentiation, but its transmitted substances between ISCs and germ cells remain identified (*Mukai et al., 2011*). The importance of gap junctions between ISCs and germ cells could also explain why ISC cellular processes are important for germ cell differentiation. Therefore, physical interactions and signaling-mediated communications between ISC cellular processes and GSC progeny likely contribute to GSC progeny differentiation collectively.

In addition, chromatin regulators are also important in ISCs to promote GSC differentiation. Eggless, a *Drosophila* H3K9 trimethyltransferase, maintains ISCs and represses *dally* and *dpp* expression in ISCs,

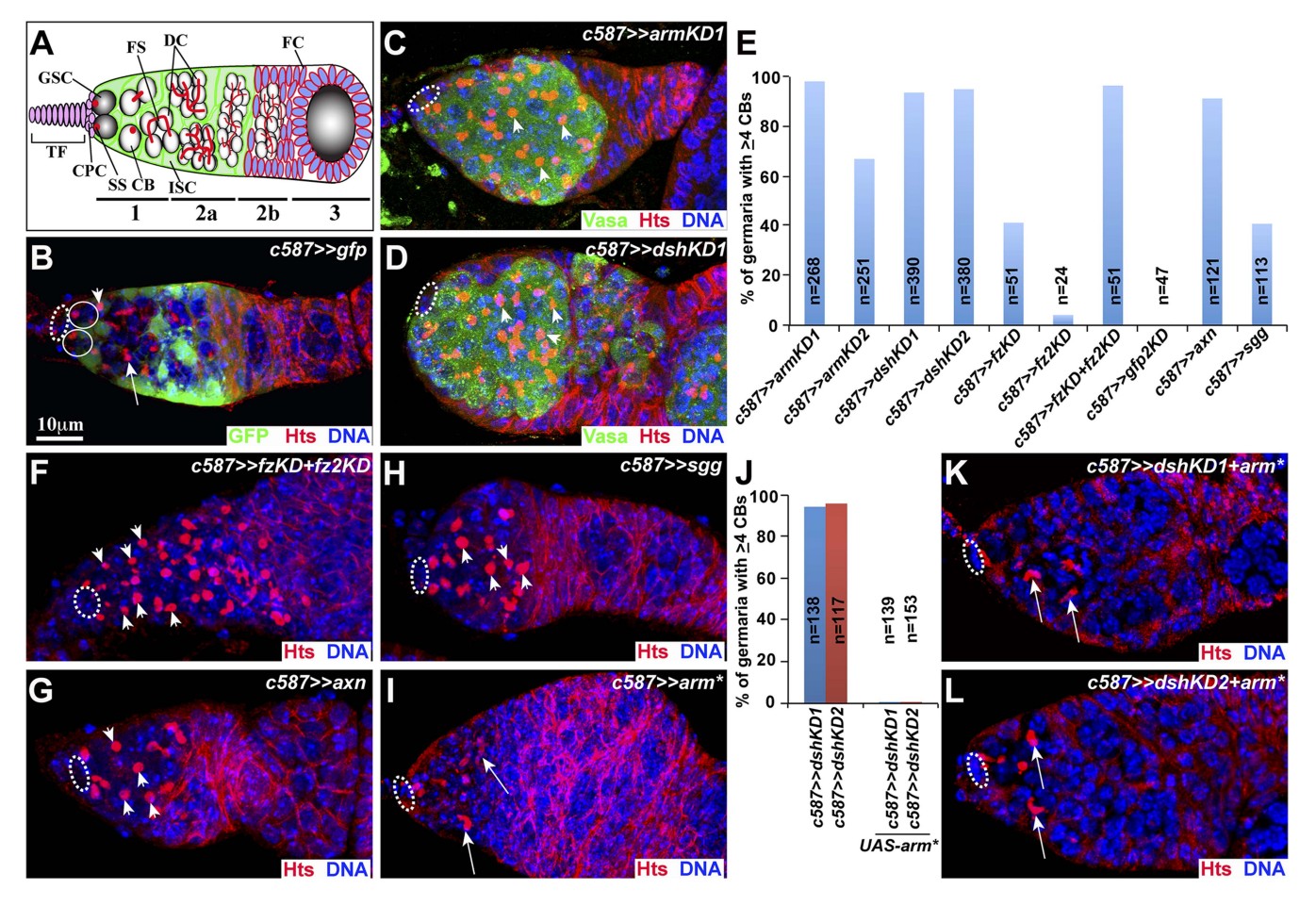

**Figure 1**. Canonical Wnt signaling in ISCs promotes germ cell differentiation. (**A**) The *Drosophila* germarium dividing into three regions 1, 2a, 2b and 3. Abbreviations: TF-terminal filament; CPC-cap cell; ISC-inner germarial sheath cell; FC-follicle cell; GSC-germ line stem cell; CB-cystoblast; DC-developing cyst; SS-spectrosome; FS-fusome. In **B–L**, cap cells are highlighted by broken ovals, whereas CBs and cysts are indicated by arrowheads and arrows, respectively. (**B**) In the *c587>>UAS-GFP* germarium containing two GSCs (spectrosomes indicated by arrowheads) close to cap cells, one CB and a few differentiated cysts are surrounded by GFP-positive ISCs. (**C–E**) In *armKD1* (**C**) *dshKD1* (**D**) germaria, many spectrosome-containing CBs accumulate far away from cap cells. (**E**) Quantification results on the percentages of the germaria exhibiting the germ cell differentiation defect (≥4 CBs). (**F–H**) *fz fz2* double knockdown (**F**), *axn*- (**G**), and *sgg*-overexpressing (**H**) germaria contain excess CBs. (**I–L**) In *arm\**-overexpressing control (**I**) and *dshKD* (**K**, **L**) germaria, GSC progeny differentiate into cysts containing a branched fusome (arrow). **J**: Quantification results.

The following figure supplement is available for figure 1:

**Figure supplement 1**. Wnt receptors FZ and FZ2 function redundantly in ISCs to promote germ cell differentiation.

thereby promoting germ cell differentiation (*Wang et al., 2011*). Similarly, Piwi functions in ISCs likely as a chromatin regulator to control germ cell differentiation partly by repressing *dpp* expression (*Jin et al., 2013*; *Ma et al., 2014*). dBre1 (a E3 ubiquitin ligase) and dSet1 (a H3K4 trimethylase) together control H3K4 trimethylation in ISCs and promote germ cell differentiation partly by limiting BMP signaling from the differentiation niche (*Xuan et al., 2013*). The potential chromatin factor without children (Woc) maintains the ISC-germ cell physical interaction via regulation of Stat-Zfh1 (*Maimon et al., 2014*). The histone demethylase Lsd1 is required in ISCs to promote germ cell differentiation by maintaining ISC survival, maintaining ISC morphology, and preventing BMP signaling from the differentiation niche (*Eliazer et al., 2011*, *2014*). Therefore, ISCs function as the differentiation niche by preventing BMP signaling and direct communication.

A recent study showed that tyrosine kinase Btk29A maintains Wnt signaling in ISCs by phosphorylating *Drosophila* β-catenin homolog Armadillo (Arm) (*Hamada-Kawaguchi et al., 2014*).

It also argues that Wnt4 activates Wnt signaling to maintain Piwi expression and repress E-cadherin expression in ISCs, thereby promoting germ cell differentiation. In contrast, this study has demonstrated that both ISC-expressing Wnt2 and Wnt4, but not Wnt4 alone, act through known Wnt pathway components to maintain active Wnt signaling, promoting germ cell differentiation. More importantly, we show that Wnt signaling is required to maintain ISCs by promoting ISC survival and proliferation. Surprisingly, Wnt signaling is dispensable for Piwi expression. Instead, we demonstrate that Wnt signaling controls the expression of four *Gst* genes to maintain the reduced redox, thereby promoting ISC maintenance and germ cell differentiation. Finally, knocking down one of the *Gst* gene, *GstD2*, in ISCs leads to the germ cell differentiation defect. Therefore, our study has revealed a novel mechanism which autocrine Wnt signaling utilizes to maintain ISCs and promote germ cell differentiation.

## Results

### Canonical Wnt signaling is required in the differentiation niche to promote germ cell differentiation

To identify the genes that function in ISCs to promote germ cell differentiation, we used *c587*-driven *UAS-RNAi* expression to knockdown individual genes in ISCs. *c587* is a GAL4 line that is specifically expressed in ISCs and early follicle cell progenitors based on *UAS-GFP* expression (*Song et al., 2004*) (*Figure 1B*). To facilitate the identification of GSCs and differentiated germ cells, spectrosomes and fusomes are labeled by Hts staining (*Lin et al., 1994*), and germ cells are visualized by Vasa staining (*Lasko and Ashburner, 1988*). In contrast to the control germarium containing 0 to 2 CBs, knocking down Wnt downstream genes *armadillo* (*arm*) and *disheveled* (*dsh*) in ISCs leads to an accumulation of many spectrosome-containing CBs (collectively referred to single germ cells at least one cell diameter away from cap cells) (*Figure 1C,D*). Based on the fact that control germaria rarely contain three CBs, the germaria containing four or more CBs are considered to exhibit a germ cell differentiation defect. Over 90% of *arm* and *dsh* knockdown germaria (*armKD1* and *dshKD1*) exhibit the germ cell differentiation defect (*Figure 1E*). Similarly, *c587*-driven expression of the RNAi lines against different *arm* or *dsh* sequences (*armKD2* and *dshKD2*) also generates the similar germ cell differentiation defect (*Figure 1E*). In *Drosophila*, Wnt ligands bind the receptor complex composed of Frizzled (Fz), Frizzled 2 (Fz2) and Arrow to activate Dsh and stabilize Arm, which forms a protein complex with a TCF (T Cell Factor)-like Pangolin in the nucleus to activate target gene expression, whereas Axin (Axn) and Shaggy (Sgg) negatively modulate Wnt signaling by promoting Arm degradation (*Logan and Nusse, 2004*). Simultaneous knockdown down of both *fz* and *fz2* (*fzKD+fz2KD*) in ISCs recapitulate the germ cell differentiation defect caused by either *armKD* or *dshKD*, although either *fzKD* or *fz2KD* yields a much weaker germ cell differentiation defect (*Figure 1E,F*; *Figure 1—figure supplement 1*). Consistently, overexpression of *axn* and *sgg* in ISCs also leads to the germ cell differentiation defect similar to that caused by *armKD* and *dshKD* (*Figure 1E,G,H*). These results indicate that Wnt signaling is required in ISCs to promote germ cell differentiation.

The expression of a constitutively active mutant *arm^{S10}* (*arm\**) leads to hyperactive Wnt signaling independently of Wnt ligands (*Pai et al., 1997*). *c587*-driven *arm\** expression alone does not cause any obvious GSC maintenance and germ cell differentiation defects, but results in severe egg chamber budding defects likely due to defective follicle cell development (*Figure 1I*). Interestingly, *c587*-mediated *arm\** expression can fully rescue the germ cell differentiation defect caused by the two independent *dshRNAi* knockdowns, indicating that Arm functions downstream of Dsh in ISCs to promote germ cell differentiation (*Figure 1J–L*). These results further suggest that the two *dshRNAi* lines are highly specific. Taken together, our findings demonstrate that canonical Wnt signaling works in ISCs to promote GSC lineage differentiation.

### Wnt signaling is required in adult ISCs to promote GSC lineage differentiation

The *c587-gal4* driver is known to be expressed in somatic precursor cells in the developing female gonad, which give rise to terminal filament, cap cells, adult ISCs, and follicular stem cells (*Zhu and Xie, 2003*). To determine if adult ISCs require active Wnt signaling for GSC lineage differentiation, we took advantage of *c587*-deriven expression of a temperature-sensitive *gal4* repressor *gal80* (*c587>>UAS-gal80^{ts}*) to allow the expression of *UAS-RNAi* lines against *dsh* and *arm* only in adult ISCs using temperature shift. At 25℃, *gal80^{ts}* is functional to prevent *gal4*-driven expression of a

*UAS* transgene, but at 29°C, *gal80^{ts}* is inactivated to allow *gal4*-driven gene expression (*McGuire et al., 2003*). After the *c587>>gal80^{ts}* control and *c587>>gal80ts UAS-dshRNAi* or *UAS-armRNAi* females eclosed at 25°C (RNAi expression is extremely low or not expressed), they either continued to be kept at 25°C for one week (keeping RNAi expression repressed) or were moved to 29°C for one week (turning on the RNAi expression due to *gal80* inactivation). For the *c587>>gal80^{ts}* control females, which were cultured at 25°C or 29°C as adults for one week, their germaria contain the normal number of CBs (*Figure 2A,B*). Interestingly, for the *c587>>gal80^{ts} UAS-dshRNAi* or

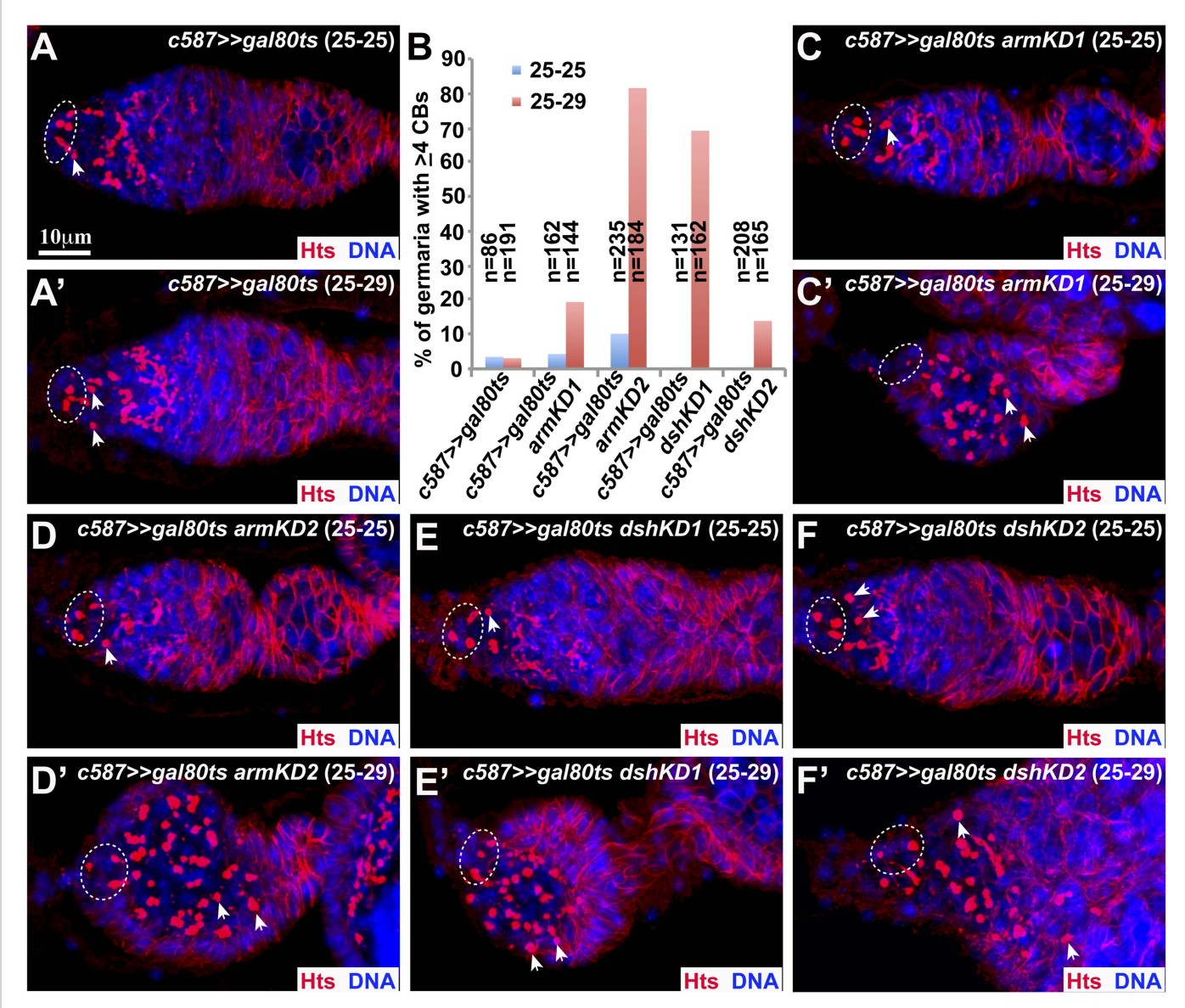

**Figure 2**. Wnt signaling is required in adult ISCs to promote germ cell differentiation. Broken ovals highlight germ line stem cells (GSCs), whereas arrowheads indicate cystoblasts (CBs). Two experimental regimens 25–25 and 25–29 mean the females cultured under 25°C or 29°C for one additional week after reaching adulthood at 25°C, respectively. (**A, A'**) Control *c587>>gal80ts* germaria contain one or two CBs under the 25–25 (**A**) or 25–29 (**A'**) condition. (**B**) Quantification results on the percentages of the germaria carrying 4 or more CBs show that adult stage-specific *arm* or *dsh* knockdown causes an accumulation of more CBs in comparison with the control. (**C–F**) *armRANi*- (**C, D**) or *dshRNAi*- (**E, F**) carrying germaria contain one or two CBs under 25–25, indicating that RNAi-mediated *arm* or *dsh* knockdown in adult inner germarial sheath cells (ISCs) under 25°C is very limited based on the germ cell differentiation phenotype. (**C'–F'**) *armRANi*- (**C', D'**) or *dshRNAi*- (**E', F'**) carrying germaria contain many more CBs under 25–29, indicating that RNAi-mediated *arm* or *dsh* knockdown under 29°C in adult ISCs leads to the severe germ cell differentiation defects.

*UAS-armRNAi* females, which were cultured at constant 25℃, their germaria contain the normal or close to normal numbers of CBs (*Figure 2B–F*). In contrast, for the *c587>>gal80^ts UAS-dshRNAi* or *UAS-armRNAi* females, which were shifted from 25℃ to 29℃ for one week, their germaria contain excess CBs, indicative of the germ cell differentiation defect (*Figure 2B,C′–F′*). These results demonstrate that Wnt signaling is required in adult ISCs to promote germ cell differentiation.

## Wnt signaling maintains ISCs by promoting proliferation and survival

Our previous studies have shown that a severe ISC loss also causes the similar germ cell differentiation defect (*Kirilly et al., 2011*; *Wang et al., 2011*). Then, we determined if Wnt signaling controls ISC maintenance by using the PZ1444 reporter to quantify ISC numbers in control and Wnt signaling-defective germaria. PZ1444 expresses nuclear LacZ in ISCs and cap cells, which can be distinguished based on nucleus size and location (*Xie and Spradling, 2000*). In the one-week-old control germaria, PZ1444 labels 30–35 ISCs in addition to cap cells (*Figure 3A,C*). In the one-week-old *arm* and *dsh* knockdown germaria, only fewer than 5 ISCs remain (*Figure 3B–D*). Some knockdown germaria have completely lost ISCs (*Figure 3B*), whereas others retain one or a few ISCs (*Figure 3D*). Consistently, overexpression of *axn* and *sgg* also leads to a severe ISC loss (*Figure 3E,F*). These results demonstrate that Wnt signaling is required to maintain ISCs.

Wnt signaling maintains ISCs possibly by promoting cell proliferation, survival, or both. Interestingly, *arm\**-expressing germaria contain significantly more ISCs (*Figure 3G*). In contrast with the one-week-old control germaria containing 32 ISCs, the one-week-old *arm\**-expressing germaria carry 130 ISCs (*Figure 3C*). Consistently, *c587*-mediated *axn* and *sgg* knockdown, which increases Wnt-signaling activity (*Heslip et al., 1997*; *Willert et al., 1999*), also leads to more ISCs (*Figure 3—figure supplement 1*). Interestingly, germ cell differentiation proceeds normally in the germaria carrying extra ISCs, suggesting that ISCs provide a permissive environment for germ cell differentiation (*Figure 3G*; *Figure 3—figure supplement 1*).

To further investigate how Wnt signaling maintains the ISC population, we examined ISC proliferation and apoptosis in the control, *armKD*, *dshKD*, and *arm\**-expressing germaria using BrdU incorporation and TUNEL (Terminal deoxynucleotidyl transferase dUTP nick end labeling)-labeling assays, respectively. In order to avoid severe ISC loss in these experiments, we purposely cultured the control, *armKD*, *dshKD*, and *arm\**-expressing females at 29℃ for shorter time than earlier experiments. BrdU incorporation labels replicating ISCs in the S-phase of the cell cycle, whereas TUNEL labeling detects fragmented DNA in dying cells. Based on BrdU-labeling results, defective Wnt signaling significantly decreases ISC proliferation, whereas hyperactive Wnt signaling significantly increases ISC proliferation (*Figure 3H–L*). Based on TUNEL-labeling results, defective Wnt signaling significantly increases ISC apoptosis, whereas hyperactive Wnt signaling significantly decreases ISC apoptosis (*Figure 3M–Q*). These results suggest that Wnt signaling maintains the ISC population by promoting proliferation and increasing survival.

## Wnt signaling is required in ISCs to prevent BMP signaling in differentiated GSC progeny and maintain long ISC cellular processes

Our previous studies have also shown that ISC loss disrupts germ cell differentiation by increasing BMP signaling (*Kirilly et al., 2011*; *Wang et al., 2011*; *Ma et al., 2014*). In the *Drosophila* germarium, BMP signaling leads to production of phosphorylated Mad (pMad) and activation of *Dad-lacZ* reporter expression in GSCs (*Chen and McKearin, 2003*; *Kai and Spradling, 2003*; *Casanueva and Ferguson, 2004*; *Song et al., 2004*). In the control germaria, pMad and *Dad-lacZ* expression is restricted to GSCs (*Figure 4A,A′,D,D′*). In the *armKD* and *dshKD* germaria, pMad and *Dad-lacZ* expression is activated in the accumulated CBs locating one to a few cells away from cap cells in addition to GSCs (*Figure 4B,B′,C,C′,E,E′,F,F′*). These results indicate that BMP signaling is spread to the differentiation zone in the Wnt signaling-defective germarium.

To investigate how Wnt signaling regulates germ cell differentiation at the molecular level, we compared the gene expression changes in fluorescence-activated cell sorting (FACS)-purified GFP-labeled control, Wnt signaling-defective *dsh* knockdown, and *axn*-overexpressing (*AxnOE*) ISCs using deep RNA sequencing (RNA-seq). Our RNA-seq results show that known *Drosophila* BMP-signaling components and regulators, including *dally* and *dpp*, which are often upregulated in defective ISCs (*Liu et al., 2010*; *Kirilly et al., 2011*; *Wang et al., 2011*; *Jin et al., 2013*; *Ma et al., 2014*), remain unchanged in the *dshKD* and

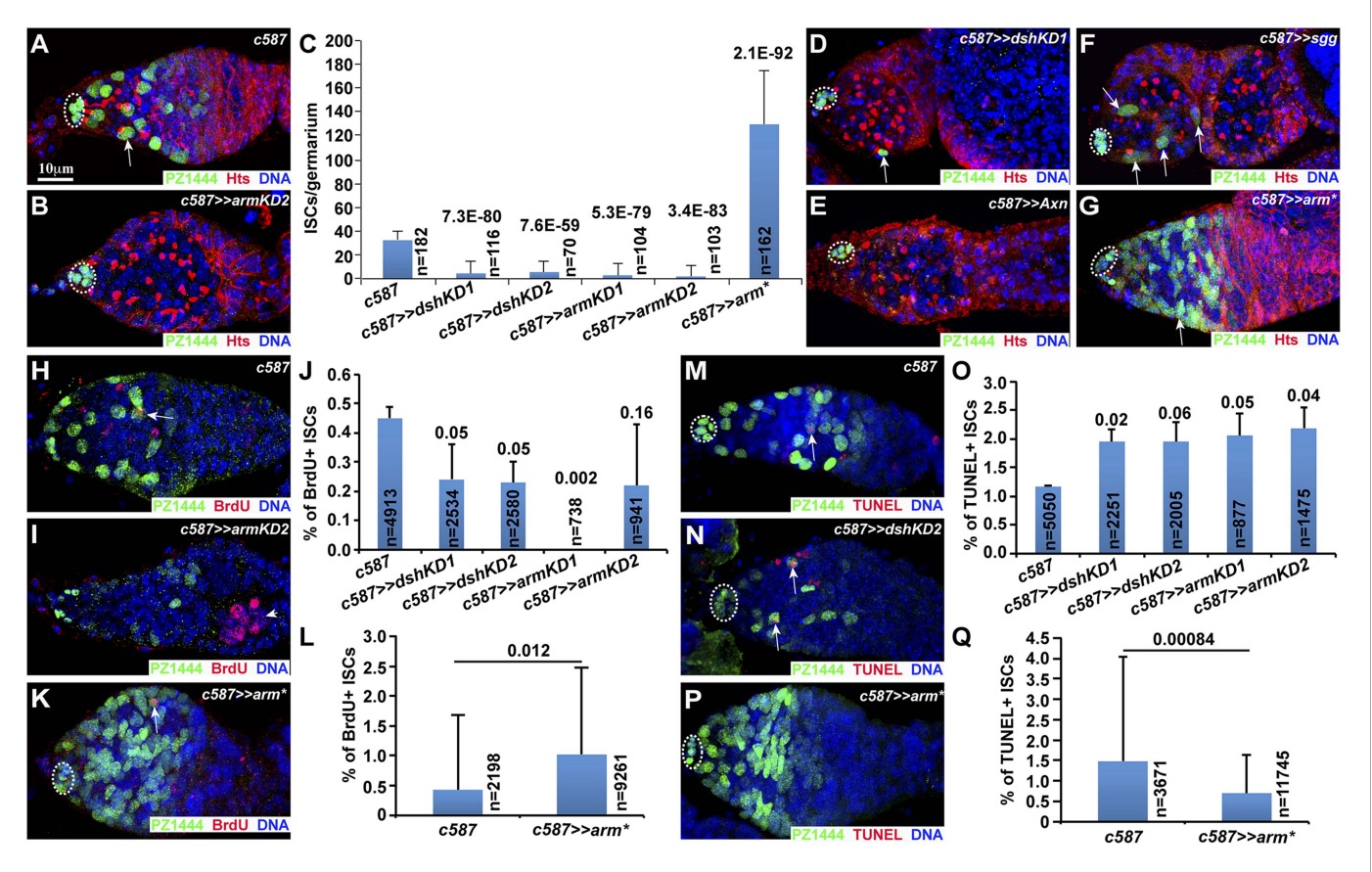

**Figure 3**. Wnt signaling maintains the differentiation niche by promoting cell proliferation and survival. Cap cells are highlighted by broken ovals. (**A**) The control germarium contains PZ1444-labeled cap cells and ISCs (one by arrow). (**B–D**) The *armKD* (**B**) and *dshKD* (**D**) germaria maintain zero and one ISC (arrow), respectively. (**C**) Quantification results on ISC numbers (mean ± standard deviation; *Student*'s *t*-test is used to calculate p values). (**E**, **F**) *c587*-directed *axn*- (**E**) and *sgg*- (**F**) overexpressing germaria contain no ISCs (**E**) and a few ISCs (**F**, arrows), respectively. (**G**) *c587*-directed *arm** expression drastically increases the ISC number (one by arrow). (**H–L**) *c587*-mediated knockdown of *arm* (**I**) and *dsh* significantly decreases BrdU-positive PZ1444-labeled ISCs (arrows), whereas *c587*-directed *arm** expression (**K**) significantly increases BrdU-labeled ISCs in comparison with the control (**H**). The arrowhead in **I** indicates a BrdU-positive germ cell cyst. **J** and **L** show quantification results. (**M–Q**) *c587*-mediated knockdown of *arm* and *dsh* (**N**) significantly increases TUNEL-positive PZ1444-labeled ISCs (arrows), whereas *c587*-directed *arm** expression (**P**) significantly decreases TUNEL-positive ISCs in comparison with the control (**M**). (**O**, **Q**) Quantification results.

The following figure supplement is available for figure 3:

**Figure supplement 1**. Hyperactive Wnt signaling increases the ISC population.

*AxnOE* ISCs in comparison with the control ISCs (*Figure 4G*). To further determine if upregulated BMP signaling is responsible for the germ cell differentiation defect resulted from defective Wnt signaling, we determined the effect of heterozygous *dpp* mutations on the differentiation defect caused by *dshKD* because they have been shown to suppress the differentiation defect caused by BMP-signaling upregulation (*Kirilly et al., 2011*; *Wang et al., 2011*). Our results indicate that the *dpp* heterozygous mutation *dpp$^{hr4}$* significantly rescues the germ cell differentiation defect caused by *dshKD1* and *dshKD2*, whereas the *dpp* heterozygous mutation *dpp$^{hr56}$* significantly rescues that caused by *dshKD1* but not *dshKD2* (*Figure 4H*). This rescue effect by the *dpp* heterozygous mutations is unlikely caused by decreasing GSCs (*Figure 4H′*). These results suggest that Wnt signaling in ISCs regulates germ cell differentiation partly by preventing BMP signaling in the differentiation niche.

Long ISC cellular processes are also required to promote germ cell differentiation (*Kirilly et al., 2011*). They can be easily visualized by *c587*-driven expression of membrane-tethered GFP, CD8GFP. In the control germaria, ISC cellular processes wrap up differentiated germ cells (*Figure 4I,I′*). In

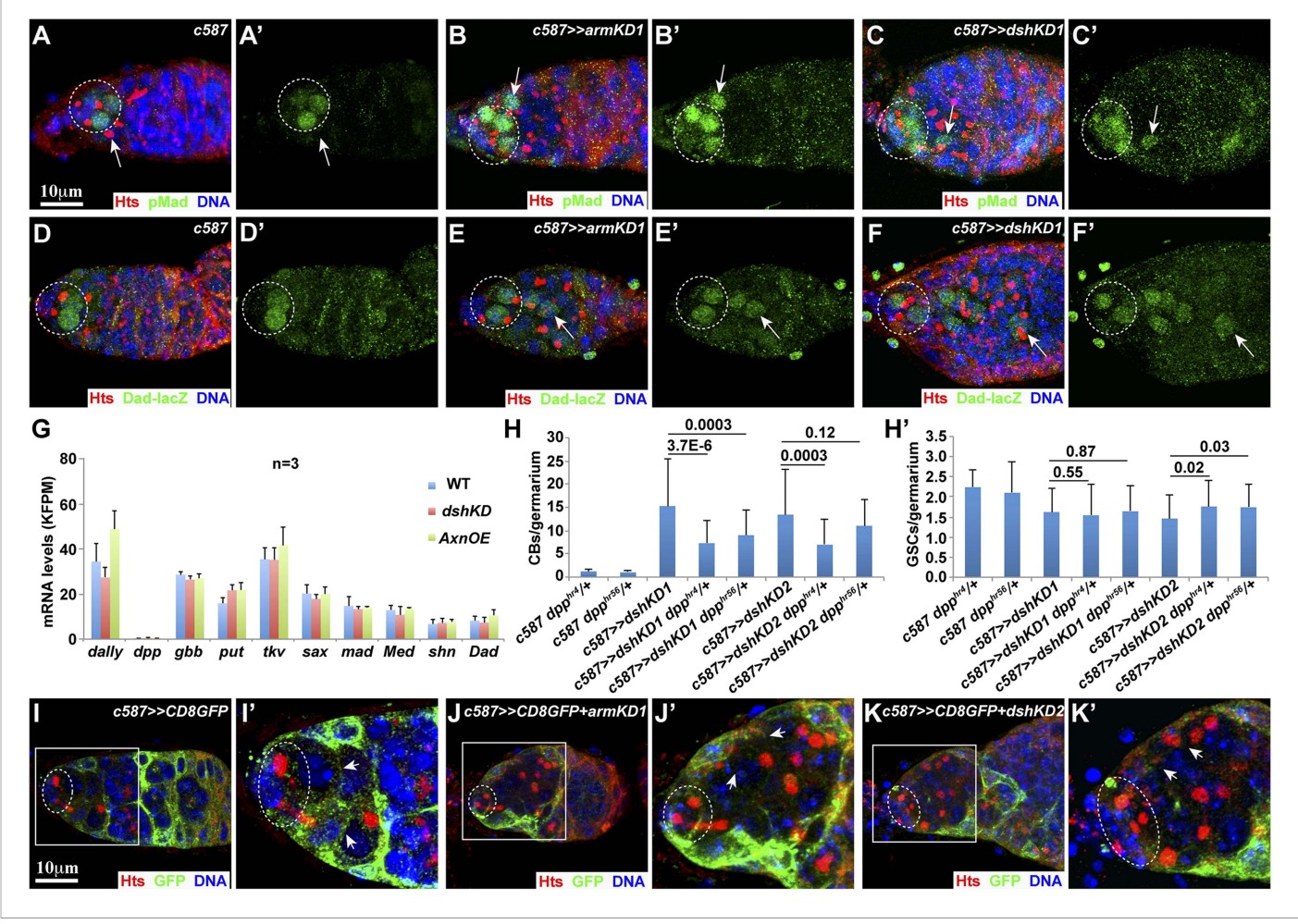

**Figure 4**. Wnt signaling in ISCs prevents BMP signaling in the differentiated germ cell zone and maintains long ISC cellular processes. Ovals indicate GSCs, whereas arrows in **A**-**F'** denote CBs or CB-like single germ cells. (**A'**–**F'**) Images only show green fluorescence of (**A**–**F**). (**A**, **A'**) In the control germarium, GSCs are positive for pMad, but one CB is negative. (**B**–**C'**) In the *armKD* (**B**, **B'**) and *dshKD* (**C**, **C'**) germaria, GSCs are pMad-positive as in the control. Although most of the accumulated CBs are negative for pMad, some CBs are pMad-positive (arrows). (**D**, **D'**) The control germarium shows that GSCs are positive for *Dad-lacZ* expression, but one CB is negative. (**E**–**F'**) In the *armKD* (**E**, **E'**) and *dshKD* (**F**, **F'**) germaria, GSCs are *Dad-lacZ*-positive as in the control. Some of the accumulated CBs are negative for *Dad-lacZ*, but the other ones exhibit low *Dad-lacZ* expression (arrows). (**G**) RNA sequencing (RNA-seq) results on the purified ISCs show that mRNA levels for the known BMP pathway components, including *dpp*, *gbb* (also encoding a BMP ligand) and *dally*, remain largely unchanged in *axn*-overexpressing ISCs or *dskKD* ISCs in comparison with the control (FKPM = reads per kilobase per million mapped reads). (**H**, **H'**) Our CB and GSC quantification results indicate that the heterozygous *dpp* mutations have partial suppression on the germ cell differentiation defects caused by *dshKD*, and the suppression is not due to the changes in GSC numbers (50 germaria examined for each genotype). (**I**–**K'**) *armKD* (**J**, **J'**) and *dshKD* (**K**, **K'**) ISCs lack their CD8GFP-positive cellular processes extending into the accumulated CBs (arrowheads) in contrast with the control ISCs extending the cellular processes wrapping up underneath germ cells (arrowheads, **I'**). (**I'**–**K'**) A higher magnification of highlighted areas in (**I**–**K**).

contrast, cellular processes in the remaining ISCs of the *armKD* and *dshKD* germaria fail to extend into the accumulated CBs (*Figure 4J–K'*). These results indicate that Wnt signaling is required to maintain ISC cellular processes.

## Wnt signaling is dispensable for maintaining Piwi protein expression in ISCs

A recent study proposes that Wnt signaling upregulates *piwi* expression in ISCs, thereby promoting germ cell differentiation (*Hamada-Kawaguchi et al., 2014*). To verify if Piwi protein is indeed downregulated in Wnt signaling-defective ISCs, we quantified the expression of Piwi protein in the

PZ1444-labeled *dshKD* or *armKD* ISCs. Piwi protein shows comparable expression levels among the examined 203 control ISCs, 48 *armKD1* ISCs, 58 *armKD2* ISCs, and 46 *dshKD2* ISCs (*Figure 5A–D*). Surprisingly, Piwi protein levels are significantly elevated in the *dshKD1* ISCs (93 examined) in comparison with the control ISCs (*Figure 5D*). In addition, we also used FACS to purify the GFP-labeled control ISCs, *axnOE* ISCs, and *dshKD1* ISCs for RNA-seq. The RNA-seq results show that the *piwi* mRNA levels are slightly upregulated in the *axnOE* ISCs and are significantly upregulated in the *dshKD* ISCs in comparison to the control ISCs (*Figure 5E*). Therefore, our results indicate that Wnt signaling promotes germ cell differentiation unlikely by maintaining Piwi expression in ISCs. However, Wnt signaling possibly works with Piwi in an unknown means to promote germ cell differentiation because the loss of their functions in ISCs leads to similar germ cell differentiation defects (*Jin et al., 2013*; *Hamada-Kawaguchi et al., 2014*; *Ma et al., 2014*).

## Wnt signaling maintains the cellular redox state in ISCs, GSCs, and early differentiated germ cells by maintaining *glutathione-S-transferase* (*Gst*) gene expression

Our RNA-seq results also show that four *Gst* genes, *GstD2*, *GstD4*, *GstD10*, and *GstE3*, decrease their mRNA expression levels significantly in both *dshKD*- and *axn*-overexpressing ISCs in comparison with the control (*Figure 6A*). GST proteins comprise a family of eukaryotic metabolic enzymes for eliminating hydrogen peroxide ($H_2O_2$) and catalyzing the conjugation of the reduced form of glutathione (GSH) to oxidized substrates for the purpose of detoxification. Thus, downregulated expression of the *Gst* genes in the Wnt signaling-defective ISCs might cause the increase in cellular reactive oxygen species (ROS), which can be detected by dihydroethidium (DHE) fluorescence. In the

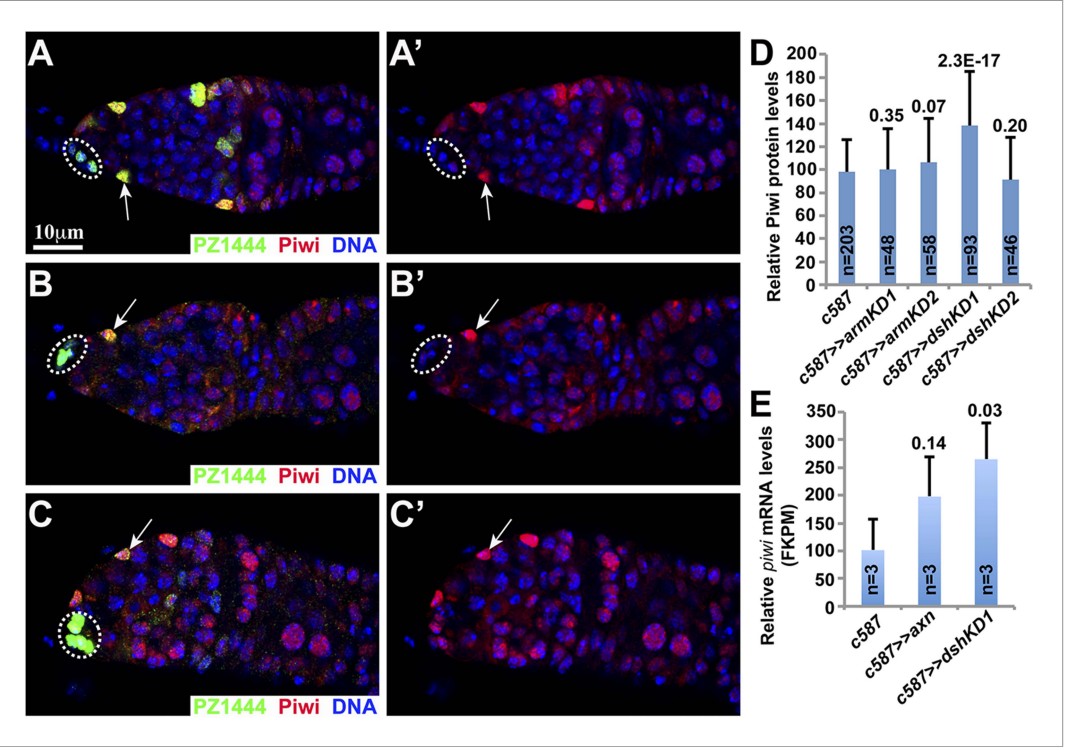

**Figure 5**. Piwi mRNA and protein expression remain normally expressed in the Wnt signaling-defective ISCs. PZ1444 is used to highlight ISCs. (**A**) The control germarium shows that ISCs (arrow) express more Piwi proteins than cap cells (oval) and underneath germ cells. (**B**, **C**) The remaining *armKD* (**B**) and *dshKD* (**C**) ISCs (arrows) still retain high Piwi protein expression. (**D**) Quantification results show that Piwi protein levels remain largely unchanged in the remaining *armKD1*, *armKD2*, and *dshKD2* ISCs in comparison with the wild-type control ISCs, but it appears to increase its expression in *dshKD1* ISCs. (**E**) RNA-seq results on the purified ISCs show that *piwi* mRNA expression levels are not changed in *axn*-overexpressing ISCs, but increase in *dskKD* ISCs, in comparison with the wild-type control ISCs.

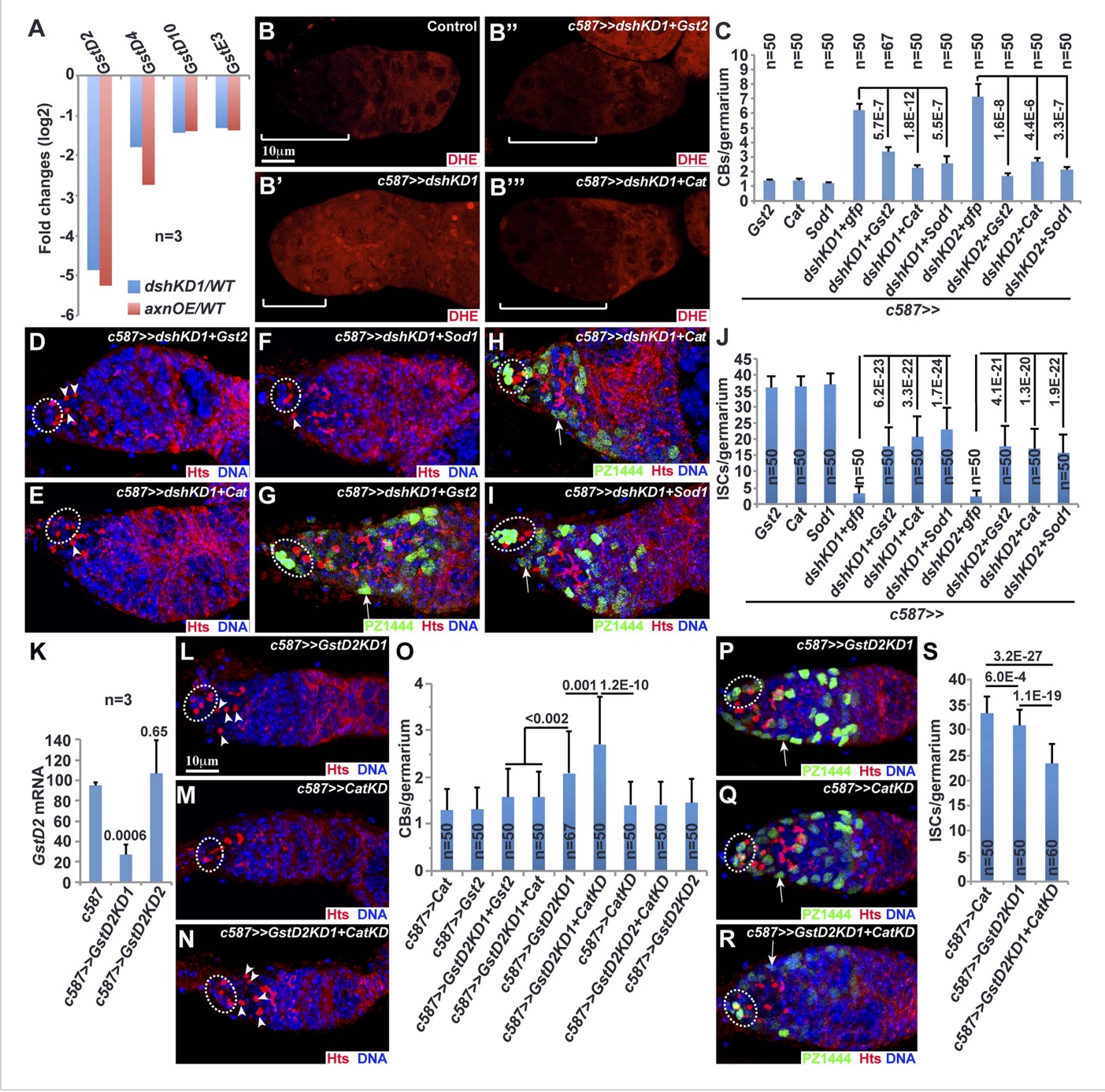

**Figure 6**. Wnt signaling maintains the reduced redox state in ISCs, promoting germ cell differentiation. (**A**) RNA-seq results of the purified ISCs show that *GstD2*, *GstD4*, *GstD10*, and *GstE3* mRNA expression levels are significantly lower in *axn*-overexpressing or *dskKD* ISCs than the control ISCs. In **B–B'''**, **D–I**, **L–N**, and **P–R**, cap cells and GSCs are highlighted by broken ovals, some CBs are indicated by arrowheads and some PZ1444-positive ISCs are denoted by arrows. (**B–B'''**) The *dshKD* germarium (**B'**) exhibits a drastic increase of dihydroethidium (DHE) fluorescence in the anterior region, including ISCs, GSCs, and early GSC progeny, in comparison with the control germarium (**B**). GST2 (**B''**) or catalase (CAT) (**B'''**) overexpression in ISCs restores low DHE fluorescence in the dshKD germariun. (**C–J**) GST2 (**D**, **G**), CAT (**E**, **H**), or superoxide dismutase1 (SOD1) (**F**, **I**) overexpression in *dshKD* ISCs significantly decreases CBs (arrowheads; **D–F**) and significantly increase ISCs (arrows; **G–I**) in comparison with *dshKD*. (**C**, **J**) Quantification results on CB and ISC numbers, respectively (for each genotype, 50 or more germaria examined). (**K**) Quantitative RT-PCR results on the purified ISCs show that *GstD2* knockdown by RNAi Line 1 (*GstD2KD1*), but not the line 2 (*GstD2KD2*), significantly decreases *GstD2* mRNA levels in comparison to the control (*c587*). (**L–O**) *GstD2KD1* (**L**) and *CatKD* germaria contain 4 CBs and 0 CB, respectively, but the *GstD2KD1 CatKD* germarium (**N**) contains 5 CBs (**O**: CB

*Figure 6. continued on next page*

*Figure 6. Continued*

quantitative results). (**P–S**) *GstD2KD1 CatKD* germarium (**R**) contains fewer ISCs (arrows) than *GstD2KD1* (**P**) and *CatKD* (**Q**) germaria (**S**: ISC quantitative results). Note: PZ1444 expression appears to be downregulated in *GstD2KD1 CatKD* ISCs.

The following figure supplements are available for figure 6:

**Figure supplement 1**. Wnt signaling maintains the reduced redox state in ISCs.

**Figure supplement 2**. Reduced redox state is important for ISC maintenance.

anterior half of control germaria, ISCs and underneath differentiating germ cells show low DHE fluorescence, indicating that these cells have low cellular ROS levels, including ISCs (*Figure 6B*; *Figure 6—figure supplement 1A,B′*). In contrast, *dshKD* and *armKD* ISCs elevate DHE fluorescence, and germ cells underneath also increase DHE fluorescence, further supporting the idea that defective Wnt signaling results in the increased cellular ROS in ISCs and their interacting germ cells (*Figure 6B′*; *Figure 6—figure supplement 1C–G*). As a distinct GST family member from the GST-D family members, GST2 has similar function in regulating the cellular redox state (*Singh et al., 2001*). Consistently, GST2 overexpression in the *dshKD* ISCs can also restore low DHE fluorescence in the anterior half of the germaria, suggesting that Wnt signaling controls the cellular redox state in ISCs, GSCs, and early GSC progeny by regulating *Gst* gene expression (*Figure 6B″*; *Figure 6—figure supplement 1H*). Catalase (CAT) can also help eliminate cellular $H_2O_2$ by converting it to free oxygen and water. As expected, CAT overexpression in the *dshKD* ISCs also restores low cellular ROS in the anterior half of the germaria (*Figure 6B′″*; *Figure 6—figure supplement 1I*). These results suggest that Wnt signaling regulates the cellular redox state in ISCs by controlling *Gst* gene expression.

Then, we determined if the restoration of the reduced redox state in *dshKD* ISCs could rescue the germ cell differentiation defect. Cytoplasmic superoxide dismutase1 (SOD1) is also important for the clearance of cellular ROS. The germaria overexpressing GST2, CAT, and SOD1 in ISCs contain normal CB numbers (*Figure 6C*; *Figure 6—figure supplement 2A–C*). GST2, CAT, and SOD1 overexpression in the *dshKD* ISCs drastically and significantly reduces CB numbers in the germaria, but does not change GSC numbers significantly, indicating that increased cellular ROS levels in *dshKD* ISCs are largely responsible for the germ cell differentiation defect (*Figure 6C–F*; *Figure 6—figure supplement 2D–G*). Similarly, GST2, CAT, and SOD1 overexpression in the *dshKD* ISCs significantly and drastically rescue the ISC number, but not to the wild-type ISC number, indicating that increased ROS is at least partly responsible for the loss of *dshKD* ISCs (*Figure 6G–J*; *Figure 6—figure supplement 2H–M*). Taken together, these results demonstrate that Wnt signaling in ISCs maintains the reduced redox state, which is partly responsible for ISC maintenance and germ cell differentiation.

### *GstD2* works with *Cat* in ISCs to control germ cell differentiation

Our RNA-seq results indicate that *GstD2* is the most abundantly expressed *Gst* genes in ISCs. In addition, it is also the most severely downregulated *Gst* gene in the Wnt signaling-defective ISCs, which prompted us to further investigate its function in promoting germ cell differentiation (*Figure 6A*). We generated two independent RNAi lines against *GstD2*, among which Line 1 (*GstD2KD1*) efficiently knocks down *GstD2* mRNA but Line 2 (*GstD2KD2*) does not based on quantitative RT-PCR results on the purified ISCs (*Figure 6K*). In contrast with the control germaria containing one CB on average, the *GstD2KD1* germaria contain two CBs, which are significantly more than the control (*Figure 6L,O*). As expected, the *GstD2KD2* germaria behave like the control, containing one CB on average (*Figure 6O*). Interestingly, *c587*-mediated expression of *Gst2* and *Cat* can also rescue the moderate germ cell differentiation defect caused by *GstD2KD1*, indicating that the differentiation defect is caused by *GstD2* knockdown in ISCs (*Figure 6O*). Considering the fact that additional three *Gst* genes are downregulated in the Wnt signaling-defective ISCs, these results indicate that *Gst* genes are required in ISCs to promote germ cell differentiation.

Both *Cat* and *Gst* genes work together to remove cellular hydrogen peroxide ($H_2O_2$). The *c587*-mediated *Cat* knockdown (*CatKD*) germaria behave like the control germaria, containing one CB on average (*Figure 6M,O*). Interestingly, *c587*-mediated *CatKD* significantly enhances the germ cell differentiation defect caused by *GstD2KD1*, and consequently the double knockdown germaria accumulate significantly more CBs than the single knockdown germaria (*Figure 6N,O*). In addition,

the *CatKD* germaria carry a normal number of ISCs, whereas the *GstD2KD* germaria carry slightly fewer ISCs (*Figure 6P,Q,S*). Consistently, the *GstD2KD1 CatKD* germaria carry significantly fewer ISCs than the *GstD1KD1* or *CatKD* germaria (*Figure 6R,S*). These results indicate that *Cat* and *Gst* genes work together in ISCs to maintain ISCs and promote germ cell differentiation, and further suggest that redox control is important for ISC maintenance.

## Autocrine Wnt2 and Wnt4 function redundantly to maintain active Wnt signaling in ISCs

Our RNA-seq results indicate that four *Wnt*-like genes are expressed in the purified ISCs, but *wingless* shows little expression (*Figure 7A*). *Wnt2* and *Wnt4* are expressed at high levels, whereas *Wnt5* and

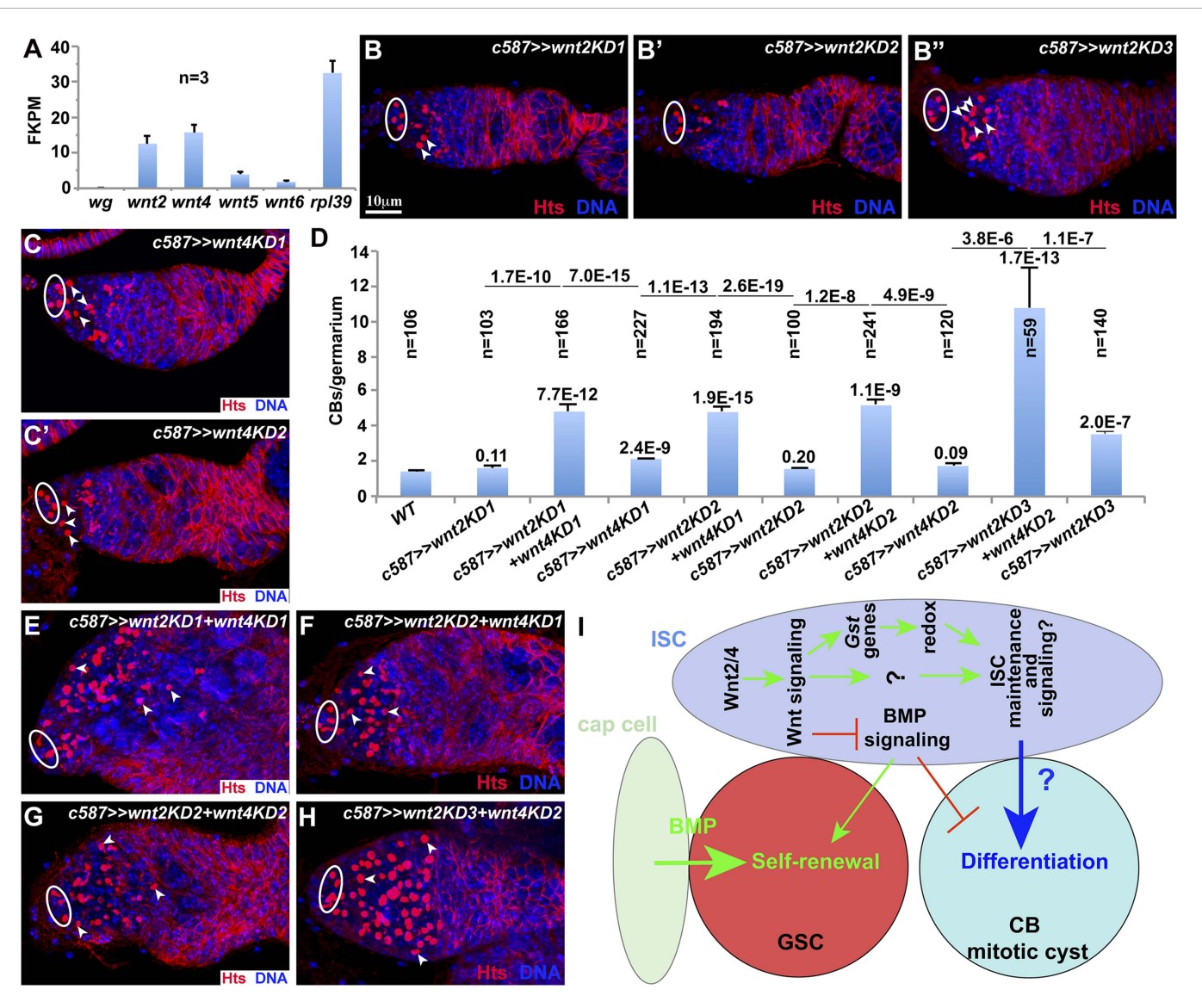

**Figure 7**. Wnt2 and Wnt4 function redundantly in ISCs to promote germ cell differentiation. (**A**) RNA-seq results show that *Wnt2* and *Wnt4* are highly expressed in the purified ISCs in comparison with *Wnt5* and *Wnt6*. In **B–C'** and **E–H**, cap cells and GSCs are highlighted by ovals, CBs are indicated by arrowheads. (**B–D**) *Wnt2* (**B–B"**) or *Wnt4* (**C, C'**) knockdown by independent RNAi lines causes a slight accumulation of CBs. (**D**) Quantification results on CB numbers (the numbers on the top of bars represent the p values in comparison with the wild-type control, whereas those on the lines indicate the p values between single and double knockdown). (**E–H**) Knocking down both *Wnt2* and *Wnt4* significantly increases CBs. (**I**) A schematic diagram showing that autocrine *Wnt2* and *Wnt4* signals control ISC maintenance and promote germ cell differentiation at least in part by maintaining the reduced redox state and preventing BMP signaling.

*Wnt6* are present in much lower levels (*Figure 7A*). A recent study has shown that *wnt4* mRNAs are indeed restricted to ISCs, whereas *wnt2* mRNAs are also present in ISCs (*Luo et al., 2015*). Knocking down *Wnt2* or *Wnt4* alone in ISCs by two or three independent RNAi lines results in no or slight germ cell differentiation defect based on CB numbers (*Figure 7B–D*). This is in contrast with the recent study claiming that Wnt4 alone in ISCs is responsible for germ cell differentiation (*Hamada-Kawaguchi et al., 2014*). Consistent with the idea that Wnt2 and Wnt4 function redundantly in ISCs, simultaneous knockdown of *Wnt2* and *Wnt4* via different combinations of RNAi lines leads to a severe germ cell differentiation defect, which is similar to that caused by *dshKD* or *armKD* (*Figure 7D–H*). These results demonstrate that Wnt2 and Wnt4 in ISCs serve as redundant autocrine signals for promoting germ cell differentiation.

## Discussion

Although the differentiation niche is critical for promoting GSC progeny differentiation, little is known about its regulation. Here, we have identified Wnt2 and Wnt4 as autocrine signals to maintain the GSC differentiation niche partly through redox regulation (*Figure 7I*). First, Wnt signaling is required in ISCs for their maintenance by promoting cell proliferation and survival. Defective Wnt signaling causes a severe ISC loss, thereby preventing germ cell differentiation. Second, Wnt signaling is required to maintain the reduced redox state by sustaining the expression of *Gst* genes. This represents a novel connection between Wnt signaling and redox control. In addition, this study has also revealed that the reduced redox state is critical for ISC survival and thus for promoting germ cell differentiation. Third, Wnt signaling in ECs promotes GSC progeny differentiation partly by repressing BMP signaling in differentiated GSC progeny. Fourth, Wnt2 and Wnt4 represent redundant autocrine signals for maintaining Wnt signaling in the germ cell differentiation niche. Wnt signaling has been shown to control stem cell self-renewal directly (*Holland et al., 2013*), whereas ROS has shown to prime hematopoietic progenitor differentiation in *Drosophila* (*Owusu-Ansah and Banerjee, 2009*). This study has demonstrated the importance of Wnt signaling in maintaining the GSC differentiation niche by reducing ROS and thus promoting GSC lineage differentiation. Therefore, this study not only has identified critical signals for maintaining the GSC differentiation niche but also has revealed a novel function of Wnt signaling in the regulation of cellular redox.

This study has demonstrated that autocrine Wnt signaling controls ISC maintenance, thereby promoting germ cell differentiation. First, canonical Wnt signaling is required in ISCs to promote germ cell differentiation. *c587*-mediated knockdown of Wnt signal transducers *Fz/fz2*, *dsh*, and *arm* as well as *c587*-directed overexpression of Wnt signaling inhibitors *sgg* and *axn* leads to similar germ cell differentiation defects. Moreover, *c587*-directed overexpression of a constitutively active form of *arm** can fully rescue the germ cell differentiation defect caused by *dsh* knockdown. Second, Wnt signaling maintains ISCs by promoting proliferation and survival. *c587*-mediated knockdown of *dsh* and *arm* leads to a severe ISC loss, whereas *c587*-mediated knockdown of *sgg* and *axn* or *c587*-directed *arm** overexpression expands the ISC population. In addition, hyperactive Wnt signaling increases ISC proliferation and decreases apoptosis, whereas Wnt signaling downregulation increases ISC apoptosis and decreases proliferation. Thus, canonical Wnt signaling maintains the differentiation niche by promoting ISC proliferation and survival. However, we could not completely rule out the possibility that defective Wnt signaling leads to the loss of adult ISCs due to early developmental defects. Third, ISC-expressing Wnt2 and Wnt4 function redundantly in the differentiation niche to promote germ cell differentiation. Our RNA-seq results show that *wnt2* and *wnt4* mRNAs are present in the purified ISCs at high levels, while other *wnt* genes are expressed at much lower levels. *c587*-mediated *wnt2* and *wnt4* double knockdown results in more severe germ cell differentiation defects than *wnt2* or *wnt4* single knockdown.

Piwi has recently been shown to be required in the differentiation niche for promoting germ cell differentiation partly by repressing *dpp* expression (*Jin et al., 2013*; *Ma et al., 2014*). Although a recent study proposes that Wnt signaling controls germ cell differentiation by regulating *piwi* expression (*Hamada-Kawaguchi et al., 2014*), this study shows that Piwi protein and mRNAs are not downregulated in Wnt signaling-defective ISCs, suggesting that Wnt signaling does not sustain *piwi* expression in the differentiation niche to promote GSC progeny differentiation. Instead, this study has further revealed that Wnt signaling maintains the differentiation niche by controlling the cellular redox. First, our RNA-seq results show that *GstD2*, *GstD4*, *GstD10*, and *GstE3* mRNA expression levels are significantly downregulated in the purified Wnt signaling-defective ISCs in comparison with

the control ISCs. Second, defective Wnt signaling in ISCs elevates ROS levels in themselves and underneath germ cells, indicating that Wnt signaling is required in the differentiation niche to maintain low ROS levels. It remains unclear if increased ROS levels in early germ cells contribute to their differentiation defects. Third, c587-directed GST2, SOD1, and CAT overexpression can dramatically suppress the ROS elevation, the germ cell differentiation retardation, and the ISC loss caused by defective Wnt signaling, indicating that ROS elevation is responsible for the germ cell differentiation defect and the ISC loss. Finally, c587-mediated knockdown of GstD2 and Cat results in the ISC loss and the germ cell differentiation defect, indicating that ROS elevation in ISCs is sufficient to cause ISC loss and retard germ cell differentiation. This study has, for the first time, demonstrated that autocrine Wnt signaling controls cellular redox state in the differentiation niche and thus promotes germ cell differentiation.

So far, various studies have revealed a number of important signaling pathways and factors in ISCs to promote germ cell differentiation by maintaining ISC cellular process-mediated germ cell–soma interaction and preventing BMP signaling (*Xie, 2013*). This study shows that Wnt signaling is required for preventing BMP signaling in differentiated germ cells and for maintaining long ISC cellular processes. Rho, Eggless, Woc, Lsd1, Piwi, EGFR signaling, and ecdysone signaling have been shown to be required in ISCs to maintain long ISC cellular processes encasing germ cells (*Schultz et al., 2002*; *Kirilly et al., 2011*; *Wang et al., 2011*; *Eliazer et al., 2014*; *Ma et al., 2014*; *Maimon et al., 2014*; *Konig and Shcherbata, 2015*). Since properly differentiated germ cells are also required to maintain long ISC cellular processes (*Kirilly et al., 2011*), it is difficult to distinguish the cause and effect of the germ cell differentiation defect and the ISC cellular process loss. In contrast, three known strategies operate in the differentiation niche to prevent BMP signaling, thereby producing a permissive environment for germ cell differentiation. First, Lsd1, Rho, and Piwi are required in ISCs to repress *dpp* mRNA expression, thereby directly preventing BMP signaling in the differentiation niche (*Eliazer et al., 2011*; *Kirilly et al., 2011*; *Jin et al., 2013*; *Ma et al., 2014*). *dpp* encodes a BMP ligand, which activates BMP signaling important for GSC self-renewal (*Xie and Spradling, 1998*). Second, Rho, Eggless, and EGFR signaling function in ISCs to repress the expression of *dally*, preventing BMP diffusion from the self-renewal niche to the differentiation niche (*Liu et al., 2010*; *Kirilly et al., 2011*; *Wang et al., 2011*). Dally, which is a proteoglycan protein facilitating BMP differentiation and signaling, is expressed in cap cells, the GSC self-renewal niche, to restrict BMP signaling to GSCs (*Akiyama et al., 2008*; *Guo and Wang, 2009*). Third, a recent study has proposed that cap cell-initiated Wnt signaling maintains the expression of *tkv* encoding a type I BMP receptor in ISCs, thereby preventing BMP signaling in differentiated germ cells (*Luo et al., 2015*). Our findings from this study have also supported the idea that autocrine Wnt signaling in ISCs promotes GSC progeny differentiation partly by repressing BMP signaling. However, our findings have also argued that inactivating Wnt signaling does not regulate the mRNA expression levels of *tkv* and other BMP pathway components in ISCs. Therefore, future research will be needed to investigate the molecular mechanisms for Wnt signaling in the differentiation niche to prevent BMP signaling and maintain long ISC cellular processes.

## Materials and methods

The *Drosophila* stocks used in this study include: c587 (*Kirilly et al., 2011*), PZ1444 (*Kirilly et al., 2011*), UAS-SOD1 (*Pan et al., 2007*), UAS-axn (BL7225), UAS-sgg (BL6818), armRNAi (BL31304; BL31305), dshRNAi (BL31306; BL31307), axnRNAi (BL31705), sggRNAi (BL35364), Wnt2RNAi (BL36722; TH00483; TH00484), and Wnt4RNAi (BL29442; TH00485). TH lines for Wnt2 and Wnt4 were constructed for this study according to the published procedure (*Ni et al., 2011*); the coding region of Gst2 and Cat were cloned into a UASp vector to generate UAS-Gst2 and UAS-Cat using standard molecular biology techniques. *Drosophila* strains were maintained and crossed at room temperature on standard cornmeal/molasses/agar media unless specified. To maximize the RNAi-mediated knockdown effect, newly eclosed flies were cultured at 29°C for a week before the analysis of ovarian phenotypes.

BrdU incorporation, TUNEL labeling, and DHE staining were performed according to the published procedures (*Xie and Spradling, 1998*; *Owusu-Ansah and Banerjee, 2009*; *Wang et al., 2011*). Immunohistochemistry was performed according to our previously published procedures (*Xie and Spradling, 1998*). The following antibodies were used in this study: rabbit polyclonal anti-β-galactosidase antibody (1:100, Cappel), mouse monoclonal anti-Hts antibody (1:50, DSHB), Guinea

pig polyclonal anti-Piwi antibodies (1:100; provided by H. Lin), rabbit monoclonal anti-pS423/425 Smad3 antibody (1:100, Epitomics), and rat monoclonal anti-Vasa antibody (1:50, DSHB). All images were taken with a Leica TCS SP5 confocal microscope.

The GFP-positive *dshKD* or *axn*-overexpressing ISCs were sorted by FACS, and mRNA isolation was carried out according to the published procedure (*Ma et al., 2014*). Following manufacturer's directions and using the Illumina TruSeq Stranded mRNA LT Kit (Illumina, MA, United States; Cat. No. RS-122-2101/2), short fragment libraries were constructed. The resulting libraries were purified using Agencourt AMPure XP system (Backman Coulter, CA, United States; Cat. No. A63880) and were then quantified using a LabChip® GX (PerkinElmer, MA, United States) and a Qubit Fluorometer (Life Technologies, CA, United States). All libraries were pooled, re-quantified, and run as 50 bp single-end lanes on an Illumina HiSeq 2500 instrument, using HiSeq Control Software 2.0.5 and Real-Time Analysis (RTA) version 1.17.20.0. Secondary Analysis version CASAVA-1.8.2 was run to demultiplex reads and generate FASTQ files.

## Acknowledgements

We would like to thank Dr H Lin, Developmental Studies Hybridoma Bank and Bloomington *Drosophila* Stock Center for reagents; the Xie laboratory members for stimulating discussions; Drs N Sachan and D Chao for critical comments on the manuscript. This work was supported by the Stowers Institute for Medical Research (TX).

## Additional information

### Funding

| Funder | Grant reference | Author |
|---|---|---|
| Stowers Institute for Medical Research | The SIMR Fund | Ting Xie |

The funder had no role in study design, data collection and interpretation, or the decision to submit the work for publication.

### Author contributions

SW, Conception and design, Acquisition of data, Analysis and interpretation of data, Drafting or revising the article; YG, Acquisition of data, Analysis and interpretation of data; XS, XM, XZ, YM, KEM, PA, JP, JH, Acquisition of data; ZY, JN, Contributed unpublished essential data or reagent; HL, Analysis and interpretation of data; TX, Conception and design, Analysis and interpretation of data, Drafting or revising the article

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
