## [Decision Letter]

Thank you for submitting your work entitled “Wnt Signaling-Mediated Redox Regulation Maintains the Germline Stem Cell Differentiation Niche” for peer review at *eLife*. Your submission has been favorably evaluated by Fiona Watt (Senior Editor), a Reviewing Editor, and two reviewers.

The reviewers have discussed the reviews with one another and the Reviewing Editor has drafted this decision to help you prepare a revised submission.

In this manuscript, Wang et al. address the role of somatic inner germarial sheath cells (ISCs) in promoting differentiation of germline cells displaced from the stem cell niche. Using a battery of genetic tools and molecular readouts, the Wnt signaling pathway is identified as a critical regulator of germ cell differentiation that acts, at least in part, autonomously within the ISCs. Clear evidence supports the conclusion that autonomous Wnt signaling is important for the viability of ISCs and that elevated ROS in Wnt signaling mutants contributes to the reduced viability of these cells. Conditions that lead to reduced ISC number led to accumulation of less differentiated germ cells, suggesting that Wnt-mediated maintenance of ISCs is important for controlling germ cell differentiation. The work adds a new mechanism to a developing field of study focused on understanding how individual signaling pathways that mediate interactions between ISC cells and germ cells influence germ cell biology.

Essential revisions:

The reviewers are unanimous in their appraisal of the manuscript and agree that if all the conclusions in the manuscript are properly supported by experiments, then this work will generate a really interesting new model for the signaling that controls somatic cell-germ cell interactions during germ cell differentiation. There are aspects of the analysis that are not conclusive, and these need to be fully addressed in order for a resubmission to be considered for publication.

The two central messages in this paper are:

1) *Wnt2* and *Wnt4* work redundantly and Wnt signaling in the ISC is somehow linked in a non-autonomous fashion to an IGS to germ cell signal, that contrary to previous reports is not *piwi* dependent.

2) The Wnt signal controls ROS signaling in the ISC cells and this is what provides the ISC to germ cell connection.

Of these two messages, the first one is a high-point of the analysis and we suggest some changes that will make the manuscript better. However, the equally important point 2 is not convincingly demonstrated. It is critical that these are addressed.

Point 1:

a) The role of *Wnt4* in the process was previously known, but the redundancy with *Wnt2* was not. On this point, rather than place the emphasis on what another laboratory “got wrong”, the authors could seize this opportunity to explain the general problem with redundancy and how this manuscript managed to avoid those pitfalls.

b) One assumes that the p-value resulting from loss of *Wnt4* alone (using D1), in Figure 7 is not correct, certainly looks that way from the graph, but if somehow this p value is in the e-9 range (rather that about 0.1 or so), then we have a big problem with the redundancy argument.

c) The reviewers feel that the dismissal of the published *piwi* data needs some substantiation. While the authors' assurance that this is due to the use of whole ovaries not accounting for ISC loss seems reasonable, the authors did not discuss an alternative signal, and they need to address this issue more directly.

Genetic experiments demonstrating that *piwi* expression in ISCs (or possibly terminal filament and cap cells given the broad expression of the Gal4 driver utilized) can rescue Btk29 mutant defects in germ cell differentiation clearly indicate an important role for *piwi* in this process (Hamada-Kawaguchi). An explanation for the ability of *piwi* expression to rescue the effects of Btk29 mutation on germ cell differentiation and Bam expression should be considered.

In addition, the possibility that loss of interactions between ISC projections and differentiated germ cells (described in [16]) upon dissociation of the germarium tissue for FACS sorting might alter the in vivo effects of genetic mutations on *piwi* expression should be carefully considered.

Finally, the *armKD* image shown in Figure 5 appears to have reduced *piwi* expression in many of the ISC cells, and the methods for quantifying Piwi protein levels are not clear. Rigorous exclusion of *piwi* as relevant for Wnt-mediated germ cell differentiation regulation would require an experiment in which *piwi* is expressed in ISCs lacking *arm* or *dsh* and assessing its ability to rescue the defects observed. The main point is that genetic evidence supports roles within ISCs for all of the above genes, and the manuscript would be enhanced by the proposal of an integrated model that considers the individual roles of each pathway in this process and how these coordinate to control germ cell differentiation.

In this context, the authors should comment on how the entire network of genes such Btk29, EGFR, Rho, Piwi, and BMP that are key factors for controlling the ISC-germ cell interactions that promote germ cell differentiation, integrates into the Wnt model. This can be addressed by textual changes only if the authors think this will enhance the paper.

Some of the Gal4 drivers used in this study are not cell type specific. It will be important to use Cap and Follicle cell drivers to confirm that the Wnt effects as stated are indeed specific to the ISC. This is an important caveat to address.

Point 2:

The ROS connection to Wnt signaling needs to be addressed with better data.

a) It is not clear, from the DHE staining shown, that knockdown of *dsh* affects ROS levels in ISCs. For example, the difference in DHE staining in 6B vs 6B' could be because the ISC cells are absent in a *dshKD* context (as suggested elsewhere in the paper). So this does not definitively establish that loss of Wnt signaling affects ROS levels in ISC cells.

b) The RNAseq data are supportive and correlative, but on its own, it is not enough to establish causality. To test whether the decrease in scavengers is the cause of the *dshKD* phenotype, it would be important to determine whether knockdown of scavengers also causes a loss of ISCs, as knockdown of *dsh* does.

c) The proliferation and apoptosis phenotypes described in Figure 3 may be due to effects during development, which would call into question whether Wnt promotes survival and proliferation during adulthood.

Here are some suggestions from the reviewers, but if the authors think of better experiments to substitute for these that will convince the reviewers, then this will be OK:

i) Controls with scavenger knockdown alone to show the DHE staining works as expected, and counter staining (e.g. LacZ in PZ1444) to identify IGS cells and show at cellular resolution the level of DHE staining in wt vs *dshKD* conditions. Since it seems all the ISC cells are gone by 7 days after heat shift, it may be necessary to either look earlier, before all the ISC cells have disappeared, or prevent apoptosis by some other means, such as overexpression of p35. If they find that DHE staining is actually higher in *dshKD* ISC cells vs wt, this would go a long way toward addressing the reviewers' concerns.

ii) Quantification of ISC number in *GSTD2 + Cat* double KD will determine if the ISC number changes or remains the same. The result needs to fit the model.

iii) To determine whether the proliferation and apoptosis phenotypes described in Figure 3 are due to effects during development, the authors should use *tsGal80* to suppress expression until adulthood, as in Figure 2, and perform TUNEL staining of *dshKD* ISC cells as they did for *arm**.

iv) Does germ-cell lethality revert when ROS scavengers are expressed in them?

[Editors' note: further revisions were requested prior to acceptance, as described below.]

Thank you for resubmitting your work entitled “Wnt Signaling-Mediated Redox Regulation Maintains the Germline Stem Cell Differentiation Niche” for further consideration at *eLife*. Your revised article has been favorably evaluated by Fiona Watt (Senior Editor), a Reviewing Editor, and two reviewers. The manuscript has been improved but there are some remaining issues that need to be addressed before acceptance, as outlined below:

1) An integrated model of how Wnt signaling and BMP signaling might work together is confusing, more so in this version of the manuscript than the last. pMAD and *Dad-lacZ* are dramatically elevated in undifferentiated germ cells in germaria with reduced *dsh* or *arm* activity (Figure 4), but the final conclusion is that BMP signaling is not affected by altered Wnt because no changes in transcript levels of other *dpp* effectors (including dad?) were detected and because a 50% reduction of *dpp* ligand did not alter the observed phenotypes, except in the case of the *dshKD1* background, where it appears to have rescued the phenotype quite dramatically. Given these data, the conclusion about independence of the pathways is not justified and needs to be tempered and explained better.

2) ROS induced in the germ cells acts downstream of the IGS cell Wnt signal to inhibit differentiation. Is there a rescue of normal differentiation by reducing ROS in germ cells when IGS cells still have reduced Wnt signaling? (A definitive comment on the link of Wnt signaling, ROS and differentiation will make this paper very exciting. However, if tools to test this conclusion are time consuming to put together, then adding this as a point of discussion will suffice).

3) One of our reviewers expressed concern that Wnt signaling during development might produce defective adult tissues in which proliferation and survival of ISCs is different. This can be easily addressed by repeating the proliferation and apoptosis experiments with *arm**, *armKD*, and *dshKD* (Figure 3) with *Gal80ts*. Please pick at least one (or more) genotype to test with *Gal80* to demonstrate that this is not the case. If this will involve multi generational crosses and a significant delay in revision, then please bring this up as an important caveat for the whole study.

4) Given the large effect of *Wnt4* knockdown, the redundancy argument is not strong. Also, the data should include p values for should provide p-values for (*Wnt2* vs *Wnt2+4* and *Wnt4* vs *Wnt2+4*.

5) *c587* is described in all sections of the paper as "ISC-specific”. This is not true and needs to be changed.

6) The new data on genetic alteration of ROS levels focused on germaria with 2 cystoblasts as indicative of a differentiation phenotype. In the text associated with Figures 1 and 2, it states that most germaria have 1-2 cystoblasts, and only gemaria with more than 4 Hts/spectrosome positive germaria were scored. It is not clear how only 2 cystoblasts can be accurately scored as a "differentiation defect” if most germaria have 1 or 2 cystoblasts present. Most likely, this is due to redundancy among GST genes, but it's hard to know how to compare this data to that presented in Figure 1.

---

## [Author Response]

*Point 1*:

*a) The role of* Wnt4 *in the process was previously known, but the redundancy with* Wnt2 *was not. On this point, rather than place the emphasis on what another laboratory “got wrong”, the authors could seize this opportunity to explain the general problem with redundancy and how this manuscript managed to avoid those pitfalls*.

Thank the reviewers for the comment. In the revised manuscript, we have put the emphasis on the requirement of both *Wnt2* and *Wnt4* in ISCs for maintaining the differentiation niche and promoting germ cell differentiation.

*b) One assumes that the p-value resulting from loss of* Wnt4 *alone (using D1), in*
Figure 7
*is not correct, certainly looks that way from the graph, but if somehow this p value is in the e-9 range (rather that about 0.1 or so), then we have a big problem with the redundancy argument*.

Indeed, knocking down *wnt2* or *wnt4* by one RNAi line yields significantly more CBs than the controls, indicating that both *Wnt2* and *Wnt4* are required for sustaining maximal Wnt signaling in ISCs to promote germ cell differentiation. However, simultaneous knockdown of both *wnt2* and *wnt4* in ISCs produces much severe germ cell differentiation defects, which is more than the additive effect of two single knockdowns. These results strongly argue that *Wnt2* and *Wnt4* function redundantly in ISCs to promote germ cell differentiation.

*c) The reviewers feel that the dismissal of the published* piwi *data needs some substantiation. While the authors' assurance that this is due to the use of whole ovaries not accounting for ISC loss seems reasonable, the authors did not discuss an alternative signal, and they need to address this issue more directly*.

*Genetic experiments demonstrating that* piwi *expression in ISCs (or possibly terminal filament and cap cells given the broad expression of the Gal4 driver utilized) can rescue Btk29 mutant defects in germ cell differentiation clearly indicate an important role for* piwi *in this process (Hamada-Kawaguchi). An explanation for the ability of* piwi *expression to rescue the effects of Btk29 mutation on germ cell differentiation and Bam expression should be considered*.

This reviewer might have misunderstood our argument. Our results on *piwi* mRNA and protein have clearly demonstrated that *piwi* expression is not downregulated in Wnt signaling-defective ISCs as suggested by Hamada-Kawaguchi et al. Others and we have confirmed the important function of Piwi in ISCs to promote germ cell differentiation (14; 25). Even though Piwi overexpression can “bypass” the requirement of Wnt signaling in ISCs in promoting germ cell differentiation, it does not necessarily support the Hamada-Kawaguchi model that Wnt signaling controls the expression of Piwi to promote germ cell differentiation. However, the rescue experimental results do suggest that Piwi might function downstream (though not directly regulated by Wnt signaling) or in parallel with Wnt signaling in ISCs to promote germ cell differentiation. We have re-written the part to avoid any potential confusion.

*In addition, the possibility that loss of interactions between ISC projections and differentiated germ cells (described in*
[16]*) upon dissociation of the germarium tissue for FACS sorting might alter the in vivo effects of genetic mutations on* piwi *expression should be carefully considered*.

We thank the reviewers for the interesting thought. The dissociation process is rapid, and the cell-sorting process is carried out at 4^°^C, which prevents transcriptional changes. Although we could not completely ruled out the suggested possibility, it is very unlikely because we have shown that Piwi protein levels are similar or upregulated in *armKD*/*dshKD* ISCs in INTACT germaria. Based on our results on *piwi* mRNA and protein, we are confident that defective Wnt signaling does not lead to the downregulated *piwi* expression in ISCs.

*Finally, the* armKD *image shown in*
Figure 5
*and B' appears to have reduced* piwi *expression in many of the ISC cells, and the methods for quantifying Piwi protein levels are not clear. Rigorous exclusion of* piwi *as relevant for Wnt-mediated germ cell differentiation regulation would require an experiment in which* piwi *is expressed in ISCs lacking* arm *or* dsh *and assessing its ability to rescue the defects observed. The main point is that genetic evidence supports roles within ISCs for all of the above genes, and the manuscript would be enhanced by the proposal of an integrated model that considers the individual roles of each pathway in this process and how these coordinate to control germ cell differentiation*.

For Piwi protein expression quantification, we used the Leica quantification software to select the nucleus area and measure the total fluorescence intensity. Like any experiments, there are some degrees of variations due to expression variation, staining variation or both. After measuring 203 wild-type ISCs, 106 *armKD* ISCs and 139 *dshKD* ISCs, we could not find any evidence supporting that Piwi protein levels are down-regulated in Wnt signaling-defective ISCs. These findings on *piwi* mRNA and protein levels are mutually supported. In this study, we argue that Piwi is NOT the direct target of Wnt signaling in ISCs. We also believe that Piwi and Wnt signaling are relevant with each other in controlling ISC maintenance and promoting germ cell differentiation because they have similar mutant phenotypes in ISC maintenance and germ cell differentiation.

*In this context, the authors should comment on how the entire network of genes such Btk29, EGFR, Rho, Piwi, and BMP that are key factors for controlling the ISC-germ cell interactions that promote germ cell differentiation, integrates into the Wnt model. This can be addressed by textual changes only if the authors think this will enhance the paper*.

According to the suggestion, we have done our best to add some of our thoughts on the relationships among the pathways in the Discussion though the connections are largely unknown. Hopefully, these speculations will help stimulate future research in these areas.

*Some of the Gal4 drivers used in this study are not cell type specific. It will be important to use Cap and Follicle cell drivers to confirm that the Wnt effects as stated are indeed specific to the ISC. This is an important caveat to address*.

As the reviewers pointed out, *c587* expresses GAL4 in not only in ISCs but also in early follicle precursor cells. In addition, the existing Gal4 drivers for cap cells, *bab1-gal4* and *hh-gal4*, are also expressed in ISCs at low levels. There is no good GAL4 driver, which is specifically expressed in early follicle progenitors. Therefore, it is very challenging to use various GAL4 drivers to address the caveat. A recently published study by the Yu Cai group shows that *wnt4* mRNAs are primarily present in ISCs, and *wnt2* mRNAs are also expressed in ISCs. These results are consistent with our RNA-seq and genetic knockdown results.

Point 2:

*[…] Here are some suggestions from the reviewers, but if the authors think of better experiments to substitute for these that will convince the reviewers, then this will be OK*:

i) Controls with scavenger knockdown alone to show the DHE staining works as expected, and counter staining (e.g. LacZ in PZ1444) to identify IGS cells and show at cellular resolution the level of DHE staining in wt vs dshKD conditions. Since it seems all the ISC cells are gone by 7 days after heat shift, it may be necessary to either look earlier, before all the ISC cells have disappeared, or prevent apoptosis by some other means, such as overexpression of p35. If they find that DHE staining is actually higher in dshKD ISC cells vs wt, this would go a long way toward addressing the reviewers' concerns.

According to the request, we have performed DHE staining in PZ1444-labeled *armKD* and *dshKD* germaria before most of the ISCs are lost. Our results confirm that *dshKD* or *armKD* ISCs exhibit elevated levels of DHE staining. New results have been added to Figure 6—figure supplement 1′.

*ii) Quantification of ISC number in* GSTD2 + Cat *double KD will determine if the ISC number changes or remains the same. The result needs to fit the model.*

According to the request, we have quantified ISC numbers in single and double knockdown germaria for *GSTD2* and *Cat*. Indeed, the ISC number is significantly reduced in *GSTD2 Cat* double knockdown germaria, which correlates with the germ cell differentiation defect. The new results have been added to Figure 6P-S.

*iii) To determine whether the proliferation and apoptosis phenotypes described in*
Figure 3
*are due to effects during development, the authors should use* tsGal80 *to suppress expression until adulthood, as in*
Figure 2*, and perform TUNEL staining of dshKD ISC cells as they did for* arm*.

According to the request, we have performed both BrdU and TUNEL experiments on the *dshKD* and *armKD* germaria. Our results show that knocking down *dsh* or *arm* indeed leads to significantly decreased BrdU-positive ISCs and significantly increased TUNEL-positive ISCs. Now new results have been provided in Figure 3.

iv) Does germ-cell lethality revert when ROS scavengers are expressed in them?

When *arm* and *dsh* are knocked down in ISCs, the severe germ cell differentiation defect appears, but the GSC loss is rarely observed. When ROS scavengers, including Cat, SOD and GST2, are expressed in ISCs, the germ cell differentiation defect caused by *arm* and *dsh* knockdown is drastically rescued, but GSC numbers remain largely unchanged. We have added the new results in Figure 6—figure supplement 2.

[Editors' note: further revisions were requested prior to acceptance, as described below.]

*1) An integrated model of how Wnt signaling and BMP signaling might work together is confusing, more so in this version of the manuscript than the last. pMAD and* Dad-lacZ *are dramatically elevated in undifferentiated germ cells in germaria with reduced* dsh *or* arm *activity (*Figure 4*), but the final conclusion is that BMP signaling is not affected by altered Wnt because no changes in transcript levels of other* dpp *effectors (including dad?) were detected and because a 50% reduction of* dpp *ligand did not alter the observed phenotypes, except in the case of the dshKD1 background, where it appears to have rescued the phenotype quite dramatically. Given these data, the conclusion about independence of the pathways is not justified and needs to be tempered and explained better.*

The reviewers must have misread our manuscript. We show that some accumulated CB-like cells are positive for pMad and *Dad-lacZ* expression, indicative of upregulated BMP signaling activities. Our quantitative results in Figure 4 show that the heterozygous *dpp* mutations, *dpp*^*hr4*^ and *dpp*^*hr56*^, can partially rescue the germ cell differentiation defect caused by *dshKD* and *dshKD2*, indicating “Wnt signaling in ISCs regulates germ cell differentiation partly by preventing BMP signaling in the differentiation niche.” However, we show that Wnt signaling does not regulate mRNA levels for BMP pathway components in ISCs, which might give the reviewers the impression “BMP signaling is not affected by altered Wnt (signaling)”. In the revised manuscript, we have made it clear that Wnt signaling in ISCs is required to prevent BMP signaling, and add the information to the model in Figure 7.

2) ROS induced in the germ cells acts downstream of the IGS cell Wnt signal to inhibit differentiation. Is there a rescue of normal differentiation by reducing ROS in germ cells when IGS cells still have reduced Wnt signaling? (A definitive comment on the link of Wnt signaling, ROS and differentiation will make this paper very exciting. However, if tools to test this conclusion are time consuming to put together, then adding this as a point of discussion will suffice).

That is a great question. All the existing tools cannot answer the question satisfactorily. We are in the process of generating new transgenic fly strains, allowing us to conditionally remove ROS in germ cells while Wnt signaling are still defective in IGS cells. It will take a long time to generate new transgenic fly strains and bring all the genetic elements together to accomplish this goal. We will leave this question for the future studies. In the Discussion, we have added the sentence “In the future, it will be important to determine if increased ROS levels in early germ cells contribute to their differentiation defects caused by defective Wnt signaling in ISCs”.

*3) One of our reviewers expressed concern that Wnt signaling during development might produce defective adult tissues in which proliferation and survival of ISCs is different. This can be easily addressed by repeating the proliferation and apoptosis experiments with* arm**,* armKD*, and* dshKD *(*Figure 3*) with* Gal80ts*. Please pick at least one (or more) genotype to test with* Gal80 *to demonstrate that this is not the case. If this will involve multi generational crosses and a significant delay in revision, then please bring this up as an important caveat for the whole study.*

In order to address this question, we would need at least a few months to bring five transgenes together, which also requires chromosomal recombination. Based on our existing experimental results, this is NOT a big issue. Our experimental results have shown that Wnt signaling is required in adult ISCs to promote germ cell differentiation and maintain themselves. In addition, we have gain-of-function and loss-of-function experiments to demonstrate that Wnt signaling is required to maintain ISCs by regulating proliferation and survival. Since we are not able to provide the experimental results excluding the possibility, we have added the statement “However, we could not completely rule out the possibility that defective Wnt signaling leads to the loss of adult ISCs due to their developmental defects.”

*4) Given the large effect of* Wnt4 *knockdown, the redundancy argument is not strong. Also, the data should include p values for should provide p-values for (*Wnt2 *vs* Wnt2+4 *and* Wnt4 *vs* Wnt2+4*.*

As suggested, we have added the P values for *Wnt2* versus *Wnt2+4* and *Wnt4* versus *Wnt2+4* in Figure 7. Our results have indicated that *Wnt2* and *4* double knockdowns have significantly severe germ cell differentiation defects than *Wnt2* and *Wnt4* single knockdowns, which support the redundant roles of *Wnt2* and *Wnt4*.

*5)* c587 *is described in all sections of the paper as “ISC-specific”. This is not true and needs to be changed.*

As suggested, we have modified the “ISC-specific” into “*c587*-mediated”.

*6) The new data on genetic alteration of ROS levels focused on germaria with 2 cystoblasts as indicative of a differentiation phenotype. In the text associated with*
Figures 1 and 2*, it states that most germaria have 1-2 cystoblasts, and only gemaria with more than 4 Hts/spectrosome positive germaria were scored. It is not clear how only 2 cystoblasts can be accurately scored as a “differentiation defect” if most germaria have 1 or 2 cystoblasts present. Most likely, this is due to redundancy among GST genes, but it's hard to know how to compare this data to that presented in*
Figure 1*.*

A normal germarium contains 0 to 2 CBs with an average of one CB. Due to the weak differentiation defect of *GstD2KD+CatKD*, we have quantified the exact CB number to determine if there is any germ cell differentiation defect. Our statistic analysis results allow us to conclude that there is any weak and yet significant differentiation defect. In Figure 4, *dshKD1* and *dshKD2* germaria contain 15 and 13 CBs, respectively. Using the germaria carrying 4 or more CBs as the differentiation defect is a more efficient quantification method in Figures 1 and 2.